# Enhancing the Effect of Tumor Necrosis Factor-Related Apoptosis-Inducing Ligand Signaling and Arginine Deprivation in Melanoma

**DOI:** 10.3390/ijms22147628

**Published:** 2021-07-16

**Authors:** Chunjing Wu, Min You, Dao Nguyen, Medhi Wangpaichitr, Ying-Ying Li, Lynn G. Feun, Macus T. Kuo, Niramol Savaraj

**Affiliations:** 1Department of Veterans Affairs, Miami VA Healthcare System, Research Service, Miami, FL 33125, USA; chunjing.wu@va.gov (C.W.); MWangpaichitr@med.miami.edu (M.W.); yli4@med.miami.edu (Y.-Y.L.); 2Sylvester Comprehensive Cancer Center, Miller School of Medicine, University of Miami, Miami, FL 33136, USA; min_you@hotmail.com (M.Y.); dnguyen4@med.miami.edu (D.N.); lfeun@med.miami.edu (L.G.F.); 3Department of Surgery, Cardiothoracic Surgery, Miller School of Medicine, University of Miami, Miami, FL 33136, USA; 4Department of Medicine, Hematology/Oncology, Miller School of Medicine, University of Miami, Miami, FL 33136, USA; 5Department of Translational Molecular Pathology, The University of Texas MD Anderson Cancer Center, Houston, TX 77030, USA; tienkuo@sbcglobal.net

**Keywords:** ADI-PEG20, apoptosis, arginine deprivation therapy, autophagy, melanoma, rhArg, TRAIL

## Abstract

Melanoma as a very aggressive type of cancer is still in urgent need of improved treatment. Tumor necrosis factor-related apoptosis-inducing ligand (TRAIL) and arginine deiminase (ADI-PEG20) are two of many suggested drugs for treating melanoma. Both have shown anti-tumor activities without harming normal cells. However, resistance to both drugs has also been noted. Studies on the mechanism of action of and resistance to these drugs provide multiple targets that can be utilized to increase the efficacy and overcome the resistance. As a result, combination strategies have been proposed for these drug candidates with various other agents, and achieved enhanced or synergistic anti-tumor effect. The combination of TRAIL and ADI-PEG20 as one example can greatly enhance the cytotoxicity to melanoma cells including those resistant to the single component of this combination. It is found that combination treatment generally can alter the expression of the components of cell signaling in melanoma cells to favor cell death. In this paper, the signaling of TRAIL and ADI-PEG20-induced arginine deprivation including the main mechanism of resistance to these drugs and exemplary combination strategies is discussed. Finally, factors hampering the clinical application of both drugs, current and future development to overcome these hurdles are briefly discussed.

## 1. Tumor Necrosis Factor-Related Apoptosis-Inducing Ligand (TRAIL) Signaling and Anti-Tumor Effect in Melanoma

### 1.1. General TRAIL Signaling in Melanoma

TRAIL was first discovered based on its sequence similarities to other members of the tumor necrosis factor family [1,2]. As its name suggests, TRAIL can induce apoptosis in tumor or transformed cells; however, normal cells are not susceptible to its death-inducing activity [3,4,5]. It has thus become a very attractive macromolecule for research and development with the hope that it can be a specific cancer-targeting drug with minimal side effects.

TRAIL is a transmembrane protein with extracellular, transmembrane, and intracellular domains. Its apoptosis-inducing function depends on the ligation of membrane-bound TRAIL receptors (also called death receptors) on target cells. To date, five receptors for TRAIL have been identified [6,7]. Two of them, DR4 (TRAIL-R1) and DR5 (TRAIL-R2) are genuine receptors that can initiate the death signaling upon binding of TRAIL. The rest three, DcR1, DcR2, and osteoprogeterin (OPG) are decoy receptors because they are generally not able to transduce the death signal of TRAIL ligation. DcR1 lacks the death domain which is required for transducing the death signal, while DcR2 has only a truncated death domain, and OPG is a soluble protein with lower affinity for TRAIL [8]. These decoy receptors are considered competitors for TRAIL binding and may serve as a form of protection in normal cells from the toxicity of TRAIL because they are usually expressed in various normal tissues [9]. However, the expression of decoy receptors on melanoma cells may not correlate with their sensitivity to TRAIL (see Section 1.2).

The ligands for DR4 or 5 are not restricted to cell membrane-bound TRAIL because TRAIL can also exist in soluble form which can be a cleaved product containing most of its extracellular domains [10] or secreted by some cells. Recombinant soluble TRAIL (sTRAIL) [3] has been widely used; in addition, agonistic monoclonal antibodies specifically engaging DR4 or DR5 are also in use [11,12].

TRAIL-induced signaling through its cognate receptors can be complex; for example, there is evidence suggesting that DR-mediated TRAIL signaling can lead to the activation of NFκB and result in the upregulation of anti-apoptotic proteins under some circumstances [13]. Nonetheless, a generally accepted pro-apoptotic pathway has been established. TRAIL or sTRAIL exists as a homotrimer with each monomer able to bind a death receptor; thus after ligation of TRAIL, the receptors will be brought together to form clusters. The clustered intracellular domains bring in the adaptor protein Fas-associated death domain (FADD) through the interaction of death domains present in both FADD and the intracellular domains to form a structure called death-inducing signaling complex (DISC) [14]. FADD further recruits procaspase-8 or 10 [15] to the DISC through the interaction of their common death effector domains, and caspase-8 or 10 will be activated presumably through auto-catalysis. These activated initiator caspases (8 or 10) then activate downstream effector caspase-3, 6, or 7 which ultimately leads to the completion of apoptosis. This pathway is usually known as the extrinsic apoptosis pathway, and the activation of extrinsic pathway alone can lead to apoptosis in so-called type I cells. However, the activation of extrinsic apoptosis pathway can interact with the intrinsic or mitochondrial pathway of apoptosis through a cytosolic protein Bid. This interaction is believed to start from the cleavage of Bid by activated caspase-8 (or possibly 10) to produce the truncated Bid (tBid) [16,17]; tBid then can translocate to the outer membrane of mitochondria to interact with Bcl-2/Bcl-xL, releasing their inhibition on Bax/Bak. Activated Bax/Bak initiates the intrinsic apoptosis pathway by forming pore complexes to induce mitochondrial outer membrane permeabilization (MOMP). This in turn leads to the release of cytochrome *c*, and other factors such as second mitochondrial activator of caspases (SMAC/Diablo), apoptosis-inducing factor (AIF), and endonuclease G (endoG) from mitochondria. Released cytchrome *c* combines with APAF-1 and procaspase-9 to form the apoptosome to produce the activated caspase-9. In addition to activating the effector caspase-3, 6 or 7, caspase-9 can activate more caspase-8 or 10 to form a positive feedback loop with the net effect of enhanced apoptosis. Released SMAC/Diablo can counter the anti-apoptotic effect of various inhibitors of apoptosis proteins (IAPs). Sometimes AIF and endoG released from mitochondria can cause apoptosis without the activation of caspase (caspase-independent apoptosis). Cells where the extrinsic pathway alone cannot induce apoptosis and require the participation of the intrinsic pathway to do so are called type II cells, and melanoma cells generally fall into this category [18] (see Diagram).



Diagram illustrated TRAIL induced apoptosis (extrinsic) pathway which can join intrinsic pathway through Bid. (Cyt C = cytochrome C; IAP = inhibition of apoptosis, MOMP = mitochondrial outer membrane permeabilization).

### 1.2. Resistance Mechanism to TRAIL in Melanoma

In melanoma, as well as in other types of cancer, many components along the apoptosis pathway can impart resistance, innate or acquired, to TRAIL.

It has been proposed that the relative amount of the DRs and DcRs on a cell would determine the susceptibility to TRAIL. However, in melanoma, evidence seems to indicate that the membrane-bound DcRs may not be correlated with the resistance to TRAIL [19,20]. Nonetheless, the expression levels of surface DR4 or 5 have been confirmed to be one of the main factors that can determine the resistance to TRAIL in melanoma [20,21]. In turn, many factors can influence the expression levels of DRs. The loss of DR expression can be a direct result of gene deletion [20,22], or of inability to anchor the receptors on the cell membrane [20]. One report shows the epigenetic control of DR4 expression [23] as exemplified by the finding that 5-aza-2′-deoxycytidine, a demethylating agent, can increase the surface expression of DR4 in TRAIL-resistant melanoma cell lines, conferring reversal of the resistance to TRAIL, and ectopic expression of DR4 in these cells also achieved the reversal of resistance. A more recent study reveals that gene silencing through the methylation of H3K27 (H3K27me3) may be responsible for the lack of expression of DR4 in lymphoid leukemia cells [24]; whether this is true for melanoma cells remains to be explored. Post-translational modification of the DRs can also affect the signal transduction of TRAIL. High expression of peptidyl O-glycosyltransferase GALNT14 is found in melanoma cell lines contributing to the sensitivity to TRAIL by modifying DR4 or 5 through O-glycosylation to enhance the DISC formation through clustering of these receptors [25]. Melanoma cells can use the downregulation of GALNT14 to attenuate TRAIL signaling at the receptor level. In summary, downregulation of surface DRs or dampened glycosylation of these DRs can confer resistance to TRAIL in melanoma.

Although the death receptors at cell surface are required for TRAIL-induced apoptosis, expression of DR4 or 5 on the cell surface will not guarantee the susceptibility to TRAIL. Many downstream components of apoptosis also play a crucial role in determining the fate of the cell after the ligation of TRAIL and DRs. These regulators mainly include cellular FLICE-inhibitory protein (cFLIP), IAPs, and Bcl-2 family proteins.

cFLIP can block the activation of initiator caspase-8 or 10 through the formation of another structure called apoptosis inhibitory complex (AIC) at DISC [26]. One early survey of a panel of melanoma cell lines found that the high expression of cFLIP is correlated with the resistance to TRAIL [19]. A later study using siRNA targeting individual inhibitor molecules including cFLIP clearly showed that knocking down cFLIP is able to impart sensitivity of melanoma cells to TRAIL [27]. The important inhibitory role of cFLIP in melanoma against TRAIL-induced apoptosis is further supported by a series of studies in melanoma where manipulating the expression or stability of cFLIP by various experimental means was able to change the sensitivity of melanoma cells to TRAIL accordingly [28,29,30,31,32,33].

IAPs are a group of inhibitors that can inhibit activated caspases through channeling the caspases to degradation via the proteasome pathway [34]. One member of the IAPs, ML-IAP (also known as Livin), was originally identified in the study of melanoma [35] as inducible by apoptosis stimuli including TRAIL. Elevated levels of ML-IAP are responsible for the insensitivity to TRAIL and other drugs in some melanoma cells. Upregulation of XIAP is also one of the mechanisms in TRAIL-resistant melanoma to prevent apoptosis [36], which can be counteracted by SMAC/Diablo. In some melanoma cells, knocking down XIAP or survivin (an IAP) seems to be very potent in overcoming the resistance to TRAIL [27].

The expression levels and activity of Bcl-2 family proteins can be important determinants of resistance to TRAIL in melanoma because they directly control the permeability of mitochondria that initiates the intrinsic pathway. These proteins can be further divided into three subgroups, anti-apoptotic, pro-apoptotic multidomain, and pro-apoptotic BH3-only. For example, Bid and Noxa belong to the pro-apoptotic BH3-only subgroup, and Bcl-2, Bcl-xL, Mcl-1 belong to the anti-apoptotic subgroup. Bax and Bak are in the pro-apoptotic multidomain subgroup. An earlier study revealed that knockdown of Bcl-2 in TRAIL-resistant melanoma cells could make these cells sensitive to TRAIL although the effect was not as potent as that of XIAP knockdown [27]. Another study found that IGF1 could induce resistance to TRAIL in melanoma cells by upregulating Bcl-2, Bcl-xL and survivin [37]. Mcl-1 knockdown in two resistant melanoma cell lines could re-sensitize them to TRAIL [38]. The importance of Bax in relation to Bcl-2 in the resistance to TRAIL in melanoma is supported by a series of studies [39,40,41]. Many more studies on the critical role of Bcl-2 family proteins in TRAIL resistance in melanoma can be found in a recent review [42].

TRAIL resistance may also be accounted for by the downregulation of initiator caspase-8 and 10 as reported in a study of acquired resistance to TRAIL in melanoma [21]. Intrinsic resistance to TRAIL has been linked to the downregulation of caspase-8 through the hypermethylation of its promoter in Ewing tumor [43] although this is still to be seen in melanoma.

### 1.3. Combination Strategies for TRAIL

Investigations on the resistance to TRAIL with melanoma and other cancer cells have been providing valuable information to design combination therapeutic strategies for TRAIL-resistant cancers. Many factors can contribute to the resistance of TRAIL, and these factors can be regulated by again multiple pathways. Therefore, the possibility of combinations with TRAIL can be huge; a recent review can be consulted for the categories of agents including irradiation that have been used with TRAIL in melanoma to overcome the resistance [42]. To reconcile the multitude of combination strategies with various immediate targets, the author of the review proposed that the underlying mechanism of all the enhancement of TRAIL is a cell cycle regulation step [44]. Nevertheless, most of the combination strategies can be seen as a convergence of signaling on the factors associated apoptosis. For example, histone deacetylase inhibitor suberoylanilide hydroxamic acid (SAHA) [45], interferon-beta [46], and aurora kinase inhibitor MLN 8237 [47] all can be combined with TRAIL to enhance cytotoxicity in melanoma, and these chemically different entities all can lead to the upregulation of DR5.

One interesting observation from the combination of TRAIL studies reveals that inhibition of B-RAF/MEK/ERK signaling actually decreases DR5 in melanoma cells, preventing or attenuating apoptosis induced by TRAIL, agonistic antibody against DR5, or T cells [48]. Similarly, downregulation of DR5 by B-RAF inhibitor and associated resistance to TRAIL receptor agonist in melanoma was reported by another group [49]. However, B-RAF inhibition was found to sensitize the resistant melanoma cells to TRAIL in an earlier study [50]. This discrepancy may be due to the fact that different B-RAF inhibitors were used. The earlier study used a pan-B-RAF inhibitor L-779450 while the latter two used V600E-specific inhibitor vemurafenib (PLX4032). This may caution the selection of drugs even the seemingly same pathway is considered as the target.

One unconventional combination is the creation of a fusion protein as anti-PD-L1:TRAIL because it may not actually enhance TRAIL signaling directly but need T cells and other myeloid cells to enhance the effect of TRAIL [51].

## 2. Arginine Deprivation Therapy (ADT) for Melanoma Cells

In our former investigation, we found that arginine deprivation can efficiently enhance TRAIL toxicity in melanoma cell lines which do not express argininosuccinate synthetase (ASS1). As proposed mono-therapies, ADT and TRAIL can enhance each other in fighting melanoma.

Arginine is regarded as a semi-essential amino acid because in addition to being synthesized de novo from metabolic intermediates, outside supply of arginine is required for tumor growth and under certain pathophysiological conditions such as in wound healing. Arginine can be biosynthesized in the urea cycle with ASS1 as a rate-limiting and essential enzyme. Arginine and its multiple metabolic intermediates participate in many biological processes [52,53]. Arginine is a direct building block for protein; some key proteins such as collagen and histone have high proportion of arginine residues. Arginine is the only direct source for generation of nitric oxide (NO). NO in turn assumes many important but sometimes contradicting functions especially when the anti- or pro-tumor activity of NO and its role in the regulation of immune response in tumor microenvironment are concerned. Polyamine which is made from ornithine, a metabolite of arginine, has pro-proliferation effect in tumor growth. Arginine is also used to make proline, another amino acid, and agmatine, a molecule in neurotransmission. Thus the lack of arginine can have a profound and complex effect on tumor cells. Nonetheless, there is no question that arginine deprivation has shown cytostatic and cytotoxic effects on various kinds of cancer cells including melanoma in vitro and in vivo.

ADT is based mainly on the fact that arginine is in high demand in cancer cells and that cancer cells cannot synthesize arginine from metabolic intermediates such as normal cells. It has been found that in many cancers, including melanoma, hepatocellular carcinoma, and prostate cancer, the incidence of low or negative expression of ASS1 is prevalent [54]. On the contrary, normal cells have low demand on arginine and are able to synthesize arginine if needed. This difference between cancer and normal cells makes ADT, such as TRAIL, a targeted therapy against cancer cells. Partial response or stable disease and minimal toxicity to normal tissue have been shown in a series of clinical studies.

Arginine deprivation can be achieved through several means; however, in practice, the commonly used are arginine-degrading enzymes in the form of recombinant arginine deiminase (ADI-PEG20) [55,56] or recombinant arginase I (rhArg1-PEG/rhArg1-PEG5000) [57]. These enzymes are used to degrade extracellular arginine, drastically restrict the external supply of arginine. In particular, ADI-PEG20 hydrolyzes arginine into citrulline and ammonium while rhArg1-PEG converts arginine into ornithine and urea. Some normal cells can convert ornithine back to citrulline with the enzyme ornithine transcarbamoylase (OTC), and ASS1 and argininosuccinate lyase (ASL) in normal cells can make arginine from citrulline. Just as ADI-PEG20 is against tumors without ASS1 expression, rhArg1-PEG shows its anti-tumor effect on cancer cells with negative expression of OTC. Herein, the discussion is mainly on ADI-PEG20 based ADT (see diagram).

### 2.1. Signaling of and Response to Arginine Deprivation in Melanoma

Upon arginine depletion or starvation, the mTORC1 complex will sense this nutritional defect in melanoma cells; in fact, arginine starvation has been shown to inhibit the mTORC1 signaling [58]. However, the details of how the signal of arginine availability is transmitted to mTROC1 can be quite complex [59]. It is later found that SLC38A9, a lysosomal transmembrane protein, plays a crucial role in sensing arginine [60]. SLC38A9 shares sequence similarity to other amino acid transporters and can actually transport arginine. Without this SLC38A9, mTORC1 cannot be activated. It is known that mTORC1 can repress autophagy, and inhibition of mTORC1 resulting from arginine depletion would activate autophagy, a process deeply implicated in tumorigenesis and antitumor processes.

Under ADT, autophagy generally acts as a pro-survival mechanism to counter the deprivation; it is even regarded by some as a mechanism of resistance to ADT. Autophagy is known as a mechanism to maintain the cellular amino acid levels [61], and a recent study also suggests the importance of autophagy in supplying arginine under arginine starvation [62]. In response to ADT, the initial autophagy can provide a window of time for cancer cells to cope with the dearth of arginine, and reprogram their biological machineries including developing long-lasting resistance to this deprivation. We and others have shown that autophagy is a response to ADI-mediated arginine deprivation in melanoma [63,64], prostate cancer [65], lymphoma [66], glioblastoma [67], small cell lung cancer [68], sarcoma [69], and breast cancer [70]. The induction of autophagy in melanoma was also noted with treatment of rhArg1 [71]. Thus, autophagy induction seems to be a common response among cancer cells to ADT, and autophagy actually can be targeted to enhance the anti-tumor effect of ADT in various types of cancers.

In addition, lack of arginine will stall the protein synthesis in the endoplasmic reticulum (ER), inducing ER stress and in turn, unfolded protein response (UPR) [72]. These responses are closely related to autophagy [73]. Arginine deprivation-induced ER stress and UPR have been observed in several cancer cell lines belonging to colorectal carcinoma, glioblastoma, and ovarian carcinoma before cell death could occur [74], suggesting they are relatively early responses. UPR is generally regarded as a means to alleviate ER stress, and it can contribute to the decision of cell fate, life or death.

UPR is also closely linked to the production of reactive oxygen species (ROS) [75]. Indeed, ROS have been detected in breast cancer cell lines with ASS1 deficiency under arginine deprivation [70,76]. ROS are induced or upregulated with arginine deprivation, and the excessive ROS was shown to damage mitochondria. In prostate cancer cells, the damage of mitochondria could be caused by prolonged arginine deprivation and be accompanied by the increase of ROS [77]. Interestingly, ER stress is also linked to mitochondria dysfunction through calcium ion transfer from ER to mitochondria [78]. The dysfunction of mitochondria, in turn, generates ROS; elevated ROS, if not cleared, will damage cellular components including DNA and lead to cell death.

ADI can induce cell cycle arrest in both normal and tumor cells [79,80]. However, experimental evidence for cell cycle arrest in melanoma is mainly from rhArg1-induced arginine deprivation [81]. Cell cycle arrest by rhArg1 has been described in hepatocellular carcinoma [82] and malignant pleural mesothelioma [83] as well. The cell cycle arrest may explain the anti-proliferative or cytostatic effect of ADT. More investigation may be needed to provide evidence for cell cycle arrest caused by ADI in melanoma. Likewise, the cellular senescence induced by ADI, which has been described for glioblastoma [84], is yet to be explored in melanoma.

Most of the above responses to ADT reflect the attempt of cancer cells to evade the effect of arginine deprivation. In some cases, when the deprivation is removed, the cancer cells will resume their proliferation and malignancy. However, if the deprivation sustains long enough, cell death will ensue. Several modes of cells death have been proposed concerning the cytotoxic effect of ADT. We have shown that ADI-PEG20 treatment can lead melanoma cell lines to caspase-dependent apoptosis [63,85]. This caspase-dependent apoptosis is also observed in lymphoma under arginine deprivation with ADI-PEG20 [66], pancreatic cancer [86], and leukemia [80]. However, in prostate cancer, the final outcome for cancer cells under ADI treatment is caspase-independent apoptosis [65], and the same cell death mode is reported in glioblastoma [67] and small cell lung cancer [68]. Moreover, autophagic death is noted in breast cancer as a result of damage to mitochondria [70]. Autophagic death is also reported in prostate cancer cells treated by ADI-PEG20, again associated with damaged mitochondria and elevated ROS [77]. It is worth noting that the final fate of tumor cells as a result of ADT may be a mixture of different modes of death. For example, cell death comprising both caspase-dependent and independent mode is suggested by data from Syed et al. [67] and Bean et al. [69]. It is thus postulated that the mode of cell death may be different in cancers depending on cell type.

Furthermore, anti-angiogenesis activity has been observed for ADT [87,88], and there is ample evidence to support that this activity is mediated by reduced NO production by ADT [89]. In addition, NO along with polyamine also influences the migration of cancer cells [90]. Therefore, ADT could hamper the metastasis and angiogenesis, as additional anti-tumor activities.

As ADT with arginine-degrading enzymes degrade the arginine in circulation, it would have other consequences. The production of NO may be affected this way. It was actually observed in a clinical study with ADI-PEG20 that the levels of NO were lowered in patients receiving this form of ADT [91]. As NO plays an important role in immune cells, ADT will likely to bring changes in the tumor immune microenvironment [92]. Furthermore, it is known that T cells need arginine to proliferate, and depletion of arginine with arginase is a strategy used by myeloid-derived suppressor cells (MDSCs) to suppress the anti-tumor function of T cells. It is thus has been reported, for instance, that rhArg1-PEG could induce the accumulation of MDSCs into tumor in a mouse model, and the accumulation of MDSCs is correlated with tumor growth [93]. This study also found that in vitro, the supply of citrulline in the presence of rhArg1-PEG could reverse the anti-proliferation effect of arginine deprivation on activated T cells, but in vivo, the increase of citrulline associated with arginine degradation by rhArg1-PEG could not prevent the adverse effect on T cells. Nonetheless, immune cells including T cells and stromal cells have the ability to make arginine de novo, and adaptation to arginine deprivation is possible. It was found that under arginine depletion, activated primary human T cells could upregulate the expression of ASS1 and use supplemented citrulline to support proliferation [94]. Contrary to the rhArg1-PEG study, another investigation showed that ADI-PEG20 could induce T-cell infiltration in a melanoma syngeneic mouse model, and ADI-PEG20 either alone or with anti-PD-1 could reduce the growth of tumor [92]. In addition, ADI-PEG20 alone or combined with anti-PD-L1 also had an anti-tumor effect in a colon carcinoma syngeneic mouse model. Riess et al., further argue that since the requirement of ariginine is concentrated mainly at early stimulation of T cells, the large amount of T cells already primed by tumor as infiltrating ones will likely be less affected by arginine deprivation [95]. In summary, these results suggest that the response to arginine deprivation inside the immune microenvironment is complex, and more investigations on this subject are warranted.

### 2.2. Mechanisms of Resistance to ADT

Re-expression or upregulation of ASS1 can be the main molecular mechanism of resistance in melanoma and quite a few other cancers. In one clinical study, two patients with initial ASS1-negative melanoma under ADI-PEG20 treatment were found to be ASS1-positive after tumor progression [96]. More cases of the upregulation of ASS1 after ADI-PEG20 treatment were reported in another investigation [97]. Studies using melanoma cell lines revealed that the suppression of ASS1 expression is due to Hif1α binding to the promoter region of ASS1, but this suppression can be relieved under ADT by induced downregulation of Hif1α and binding of upregulated c-Myc to replace Hif1α. However, in a cell line that cannot develop resistance, c-Myc somehow cannot replace Hif1α at the binding region [98]. Subsequently, it was found that ADI-PEG20 could activate Ras and the downstream ERK and PI3K pathways to phosphorylate c-Myc, resulting in the diversion of c-Myc from degradation by proteasome [99], and the p300-HDAC2-Sin3A system is responsible for initiating the degradation of Hif1α at the ASS1 promoter [100].

In other tumor cells as reported in lymphoma, methylation of the promoter region can be one of the mechanisms that silence the expression of ASS1, and demethylating agents such as 5-aza-2′-deoxycytidine can induce the re-expression of ASS1 and confer resistance to ADI-PEG20 [66]. This methylation mechanism is also observed in malignant pleural mesothelioma [101], glioblastoma [67], ovarian cancer [102], and myxofibrosarcoma [103]. However, so far there has been no report on demethylation and subsequent upregulation of ASS1 expression as a mechanism to acquire resistance in tumor cells to ADT.

Apart from the upregulation of ASS1, reprogramming of tumor cell metabolism may also contribute to the resistance to ADI-PEG20 in melanoma [104]. By studying induced-resistant melanoma cell lines, the investigators found that these cells had lowered mTOR activity but enhanced glycolysis in addition to upregulated ASS1 expression. More interestingly, these cells developed a shift to glutamine utilization, and c-Myc seemed to be the regulator for all the metabolic changes revealed in this study. Another study employing ADT-resistant sarcoma and melanoma cell lines identified attenuated glycolysis, and enhanced glutamine usage along with enhanced oxidative phosphorylation was identified [105]. Furthermore, this study also revealed an increase in serine biosynthesis in the resistant cells.

As ADI is of mycoplasma origin, it is quite immunogenic in humans. Although the pegylation mitigates the immunogenic issue to some extent, ADI-PEG20 remains capable of inducing an immune response resulting in the production of neutralizing antibodies. The production of such antibody has been reported in advanced melanoma [91,106] and advanced hepatocellular carcinoma [106,107] patients undergone ADI-PEG20 treatment. This confers another mode of resistance to ADI-PEG20-based therapy.

### 2.3. Combination Strategies to Enhance ADT

As a monotherapy such as TRAIL, ADT can be used in combination to enhance other anti-tumor drugs, and also benefit from the anti-tumor effect from these drugs. Although the reported combination cases are not as numerous as those for TRAIL, there are many proposed combination strategies which can potentially improve the efficacy of ADT.

As discussed before, autophagy is usually a pro-survival mechanism for tumor cells induced by ADT; therefore, inhibition of autophagy can enhance the cytotoxicity of ADT except in tumors that will undergo autophagic death [70]. Autophagy inhibitors such as chloroquine have been used in researches studying the tumor response to ADT. In fact, combination of ADT with autophagy inhibition has been investigated with promising results. An example of proposed combination strategies using chloroquine and ADI-PEG20 was conducted in sarcoma [69]; combination of rhArg1 was investigated for triple-negative breast cancers with chloroquine or 3-methyladenine [108], and in non-small cell lung cancer with chloroquine or LY294002 (not a direct autophagy inhibitor) [109]. However, the possible systemic toxicity of these inhibitors especially chloroquine to normal cells should be noted.

More strategies involve chemotherapeutic drugs. The enhancement of gemcitabine in pancreatic cancer with ADI was noted by different groups probably through multiple mechanisms including cell cycle arrest, upregulation of caspases, reduction of the ribonucleotide reductase subunit M2, etc. [110,111]. Enhancement is also achieved through combination of ADI-PEG20 with cisplatin in melanoma. ADI could induce the downregulation of proteins responsible for DNA damage sensing and repair such as FANCD2 and ATM; furthermore, cisplatin could attenuate ADI-induced autophagy and further repress the expression of ASS1 through DEC1 [112,113]. Other promising combination candidates with ADI-PEG20 include 5-flurouracil for hepatocellular carcinoma [114], docetaxel for solid malignant tumors [115], and temozolomide for glioblastoma multiforme [116].

Other drugs showed encouraging effect in combination with ADT as well. An early study using dexamethasone and ADI in leukemia CEM cells showed synergestic effect against tumor [117]. Enhancement of cell killing was found in primary gliobalstoma treated by ADI derived from *Streptoccus pyogenes* with different drugs including SAHA and Palomid 529 (an mTOR inhibitor) [118]. Panobinostat (a histone deacetylase inhibitor) has also been reported to work with ADI-PEG20 for pancreatic ductal adenocarcinoma [119]. More recently, arginine analogs have entered the scene of combination therapy with ADT. Indospicine, a natural arginine analog from plant, when combined with rhArg1 has shown enhanced anti-tumor activity in colorectal cancer cell lines [120]. Elevated ER stress and perturbation of some pro-survival pathways may be the underlying reason for this enhancement, and these induced changes and associated enhanced cell death are dependent on protein translation, suggesting that the misincorporation of the analog into newly-made protein may help to initiate cell death signaling. Arginine deprivation, as the investigators postulated, may help to enhance this misincorporation. In addition, data from this study also indicate that the combination may not be cytotoxic to normal cells.

## 3. Combination of TRAIL and ADI-PEG20 for Melanoma Cells

As one of the proposed combination strategies, we have used ADI-PEG20 with soluble TRAIL to accelerate the death of melanoma cells. By rapidly leading to cancer cell death, combination strategies not only achieve effective cytotoxicity but also reduce the likelihood of resistance, such as induced upregulation of ASS1 or production of neutralizing antibody against ADI.

In our investigation [85], melanoma cell lines (A2058, A375, Mel-1220, and SK-MEL-2) all showed response to ADI-PEG20 mono-treatment from 48 to 72 h as detected with MTS assay; live cells were from around 50% to around 70% of the controls. TRAIL alone also slightly reduced the live cells in A2058 (by around 15% vs. control) and SK-MEL-2 (by around 30% vs. control) while virtually no change was seen in A375 and Mel-1220 compared with their controls. However, the combined treatment was able to reduce the live cells by about 85% in A375 and SK-MEL-2, 90% in Mel-1220, and 95% in A2058, comparing with their respective controls. This is more significant for Mel-1220 and A2058 because Mel-1220 while sensitive to ADI-PEG20, needs relatively prolonged arginine deprivation to undergo apoptosis, and A2058 can be induced by arginine deprivation to upregulate ASS1 expression and become resistant to ADT.

PI/Annexin V double-staining flow cytometry data from A2058 and A375 showed correlated cell death with caspase activation, and inhibitor to caspase-3 could prevent part of the cell death, proving that the part of the death caused by this particular combination is caspase-dependent. ADI-PEG20 single treatment resulted in the increase of surface DR4 and 5 in both cell lines, as well as the upregulation of a pro-apoptotic protein Noxa and the downregulation of an IAP protein survivin. Single treatment by TRAIL could also upregulate the levels of Noxa although not as strong as that by ADI-PEG20. The interesting observation is that the combination can further increase the levels of Noxa to an extent that is higher than that induced by either single treatment. Similarly, a single treatment by TRAIL could induce the production of tBid, and the production is further enhanced in the combination treatment. The importance of tBid participation in causing the cell death by the combination was confirmed with siRNA-induced knockdown of Bid, where apparent inhibition of cell death was achieved under the combination treatment.

In a subsequent study in A375 and A2058 [64], we confirmed that the combination of TRAIL and arginine deprivation could result in enhanced apoptosis, and more interestingly found that autophagy induced by arginine deprivation could be attenuated by the inclusion of TRAIL. This attenuation was associated with the cleavage of at least two essential proteins, Beclin-1 and ATG5, essential for autophagosome formation (which is a key step in autophagy). This cleavage was not apparent in cells with TRAIL single-treatment, but prominent when cells were treated with arginine-depleted medium plus TRAIL. Inhibition of various caspases (3, 6, 8, 9, and 10) could achieve prevention of the cleavage of Beclin-1 and ATG5 to different extents, and the degree of prevention of cleavage is generally correlated with that of the prevention of cell death. Inhibition of single caspase in the combination treatment would bring the cell death back to a level higher than but close to that resulting from arginine deprivation single-treatment. For example, inhibition of caspase-8 resulted in 97% live cells in A375 and 96% in A2058 if their corresponding arginine deprivation single-treatment control was set to 100%. These results highlight the critical role of autophagy as a pro-survival mechanism in the initial response to ADI and the apoptosis-inducing function of TRAIL.

Taken together, this combination is an example displaying the interaction of two anti-cancer drugs facilitating each other by tweaking the cellular pathways of apoptosis and autophagy in melanoma cells. The ADI-PEG20 could make the cells more susceptible to TRAIL through upregulating DR4/5, pro-apoptotic protein Noxa, and downregulating IAP member survivin. At the same time, TRAIL can induce the production of tBid through caspase-8 or 10 activation to enhance the mitochondrial apoptosis pathway. The increase of Noxa induced somehow by TRAIL may also contribute to the mitochondrial pathway. More importantly, TRAIL through ADT-induced upregulation of surface DR4/5, can activate more caspase-8 to cleave Beclin-1 and ATG5, leading to the attenuation of autophagy; thus accelerate the transition from autophagy to apoptosis.

One important feature of combining arginine deprivation with TRAIL is that the resulting modality can still preserve the specificity to cancer cells without systemic toxicity resulting from harming the normal cells. A recent development of this combination is the production of a fusion protein comprising TRAIL and ADI. This novel fusion protein has been shown to have activity in a xenograft model of colorectal cancer [121]. According to the investigators, the fusion protein also added structural synergy between these two proteins in addition to their functional one. This may spawn the design of other novel fusion proteins with distinct functions but with the same aim to attack tumor cells.

## 4. Future Perspective and Summary

### 4.1. TRAIL

The current issue with TRAIL as an anti-tumor agent is that its activity in clinical trials has been disappointing. Reasons for this failure have been postulated. The soluble TRAIL used is not very stable, with poor pharmacokinetic property such as short half-life, and the agonist antibody only binds to one type of death receptor while the binding to receptors is not strong or good enough to start the successful signaling [122]. The soluble form is also regarded not as potent as the membrane-bound TRAIL in inducing apoptosis possibly because the former cannot bring the receptors into proper conformation [123]. Delivery of TRAIL to tumor site may also be a potential problem. Different ways to improve TRAIL have been tested and showed positive results. Tagging strategies are used to improve the stability and efficacy of soluble TRAIL as a trimer through adding protein tags to soluble TRAIL [124,125]. Mesenchymal stem cells are also engineered to express membrane-bound TRAIL in its natural form. The exosomes derived from such cells, or the cells themselves have been used to deliver TRAIL to tumor [126,127,128,129]. These strategies not only provide the natural form of TRAIL, but also take advantage of the special “homing” properties of mesenchymal stem cells to deliver TRAIL to tumor sites. As viral vectors have been used to deliver TRAIL [130], nanoparticles are also adopted for this purpose [131]. Second-generation TRAIL receptor angonists such as hvTRA [49] or IZI1551 with IAP antagonist [132] have also showed improvement and enhancement, respectively. It is expected that improving the TRAIL or agonist molecule itself and the delivery system with various means would lead to a new generation of TRAIL with much better biological and pharmacological properties than those used in previous trials.

### 4.2. ADI-PEG20

As induced upregulation of ASS1 can be the main cause of resistance to ADT in melanoma, the convenient way to overcome this is to target the transcription factors c-Myc that contribute to this upregulation as reported in the study of combining cisplatin with ADI-PEG20 [112]. Inhibition of the pathways that stabilize c-Myc under arginine depletion can also be considered [99]. One relatively indirect way is to target the metabolic changes accompanying the resistance [104]. Apart from the induction of ASS1 expression in target cancer cells, one major obstacle to the clinical application of ADI-PEG20, especially when used as a mono-therapy, is its immunogenicity. An attempt to reduce the immunogenicity of ADI from *Mycoplasma hominis* through the removal of B-cell epitope was reported recently [133]; it is a computational approach to find antigenic residues and replace them with other amino acids while keeping the structural stability of the enzyme. A mutant protein with less immune-reactivity was generated computationally; unfortunately, its enzymatic activity and immunogenicity still need to be confirmed in the laboratory. Notwithstanding, modification of ADI to reduce its immunogenicity while keep or even improve its activity and stability can be worthwhile for the future development of ADI-PEG20-based therapy. Another way to avoid the immunogenicity of ADI is to use rhArg1-PEG5000 since it is of human origin. Although rhArg1-PEG5000 compared with the natural arginase has improvement such as higher affinity for arginine and longer half-life, there is still an innate issue associated with arginase, that is, the degradation product ornithine. It is known that only specific tissue can re-utilize ornithine [134], and the accumulation of ornithine in normal tissue as a result of rhArg1-PEG5000 may pose as toxic. To address this issue, novel combination strategies may be needed. In short, ADT will benefit from the continuing development of arginine-degrading enzymes to resolve issues that have hampered its clinical application.

In summary, combination therapy can be an effective modality not only to increase cytotoxicity to cancer cells through the synergy of anti-tumor activities between the two agents, but also reduces the likelihood of developing drug resistance. It would be ideal to combine two well-investigated and approved drugs, but new agents should be explored for obvious reasons. The designing of such strategies largely depends on our understanding of the mechanisms concerning the resistance development in the target cancer cells and the action of these drugs. Then the selection of the combination partner can be based on its action on the resistance mechanism to the other drug or the signaling that cross-talk with the action of the other drug. In the case of TRAIL, the combination candidates are often those which can ultimately bring changes to the expression of the factors in the apoptosis signaling pathways, reflecting the innate apoptosis-causing function of TRAIL and the usual tendency of tumors to evade apoptosis. For ADI-PEG20, the other partner of the combination may be more likely an agent targeting the metabolic response such as autophagy and cellular stress. With the accumulation of researches on combination therapy against cancer, more targets and biomarkers will be discovered in melanoma and other cancers, and new combination strategies (which may not be limited to only two partners) can be designed for TRAIL or ADI-PEG20. Existing combination strategies can be further refined and improved. It is quite hopeful that new modalities based on TRAIL or ADT will be derived from these combination strategies to benefit melanoma or other cancer patients.

## Data Availability

All data are provided in the manuscript.

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
