# Peer review of "Enhancing the Effect of Tumor Necrosis Factor-Related Apoptosis-Inducing Ligand Signaling and Arginine Deprivation in Melanoma"

_ijms, 2021, doi:10.3390/ijms22147628_

Round 1

Reviewer 1 Report

The authors composed a good overview of the current understanding of TRAIL mediated apoptosis as well as arginine deprivation mediated effects. The authors introduced apoptosis signaling and highlighted potential links to improve TRAIL mediated apoptosis by utilizing arginine deprivation.

Though in general the manuscript is reasonably well written several severe spelling mistakes were made.  For example page 3: O-glycolylation; or IPAs; TRAIL signaling the at the receptor level... Page 7: stiumaltion ... Page 3: caspase-8 or10 (missing space).. to undergo apoptosis are (not is) called type II cells...  Page 7: It is thus has been reported,... that the since the ... Page 10: Deliverty.. deliver TRAIL to tumor cells...

So a careful reading is required prior to publishing the manuscript.

The authors described apoptosis signaling with all key mediators, however, the part where pro-apoptotic Bcl-2 proteins (Page 2) lead to the release of cyt c was missing/short, particular the critical step of MOMP (mitochondrial outer membrane permeabilization) was not described. This needs to be added. The loss of mitochondrial membrane potential is a key effect that is used for analyzing the activation of intrinsic apoptosis signaling and thus is important.

Some statements I find worth of checking again to see if that is the best way to present the information.

Page 3: cFLIP is mediating its inhibitory effect by replacing caspase-8 or -10 (initiator caspases) in the DISC...

Page 3: IAPs pose as a group of... IAPs are a group inhibitors

Page 6: ... it would have a systemic consequence. Its not really clear what the authors mean by this statement.

On page 9 the authors state that TRAIL can provide tBid, which is of course not the best way to describe the process of capsase-8 mediated cleavage of Bid into truncated Bid (tBid).  This should be rephrased. Later in that sentence the authors indicate that TRAIL signaling via tBid increases Noxa. It is unclear if the authors mean that TRAIL increases Noxa by inducing transcription or because Noxa is released from the binding to anti-apoptotic Bcl-2 proteins? 

At page 10 the authors state that new generations of TRAIL could have much better drug-like properties... This is a bit confusing and should be explained a bit more. What properties do drugs have that TRAIL lacks? 

Author Response

We have corrected the above mistakes, and we also read the manuscript again and corrected some more similar mistakes.

The authors described apoptosis signaling with all key mediators, however, the part where pro-apoptotic Bcl-2 proteins (Page 2) lead to the release of cyt c was missing/short, particular the critical step of MOMP (mitochondrial outer membrane permeabilization) was not described. This needs to be added. The loss of mitochondrial membrane potential is a key effect that is used for analyzing the activation of intrinsic apoptosis signaling and thus is important.

The critical MOMP has been added in the description of the intrinsic apoptosis pathway.

Some statements I find worth of checking again to see if that is the best way to present the information.

Page 3: cFLIP is mediating its inhibitory effect by replacing caspase-8 or -10 (initiator caspases) in the DISC...

We rewrote the sentence to indicate that cFLIP will form an apoptosis inhibitory complex to block the activation of caspase-8 or 10 according to the cited reference.

Page 3: IAPs pose as a group of... IAPs are a group inhibitors

This sentence has been rewritten as suggested by the reviewer.

Page 6: ... it would have a systemic consequence. Its not really clear what the authors mean by this statement.

We changed “a systemic consequence” to “other consequences” to make the meaning clearer.

On page 9 the authors state that TRAIL can provide tBid, which is of course not the best way to describe the process of capsase-8 mediated cleavage of Bid into truncated Bid (tBid).  This should be rephrased. Later in that sentence the authors indicate that TRAIL signaling via tBid increases Noxa. It is unclear if the authors mean that TRAIL increases Noxa by inducing transcription or because Noxa is released from the binding to anti-apoptotic Bcl-2 proteins? 

We rewrote this sentence to reflect the fact that TRAIL induces tBid through the activation of caspase-8 or 10. The latter part of the sentence concerning Noxa was removed, and a new sentence was added to indicate that TRAIL could induce the increase of Noxa in our observation while the mechanism of this increase was not clear.

At page 10 the authors state that new generations of TRAIL could have much better drug-like properties... This is a bit confusing and should be explained a bit more. What properties do drugs have that TRAIL lacks? 

We agree that the word “drug-like” is confusing here, and we changed that to “biological and pharmacological”.

Reviewer 2 Report

The review by Wu et al on the combination of TRAIL and ADT in Melanoma gives a comprehensive analysis on the biological and biochemical rationale of combining Arginine deprivation with targeting TRAIL to combat melanoma. The manuscript is well written and interesting to read. Many details are given in the various effects of how Arginine starvation may increase a tumors sensitivity to TRAIL signaling. In this context, however, it would be good to provide some more information about the clinical application of TRAIL-agonists. Hence I would wish for a short paragraph on previous and ongoing clinical trials with TRAIL agonists.

Minor:

There seems to be a mistake in the abbreviations: ADI-PEG20 appears twice.

Author Response

The review by Wu et al on the combination of TRAIL and ADT in Melanoma gives a comprehensive analysis on the biological and biochemical rationale of combining Arginine deprivation with targeting TRAIL to combat melanoma. The manuscript is well written and interesting to read. Many details are given in the various effects of how Arginine starvation may increase a tumors sensitivity to TRAIL signaling. In this context, however, it would be good to provide some more information about the clinical application of TRAIL-agonists. Hence I would wish for a short paragraph on previous and ongoing clinical trials with TRAIL agonists.

Recombinant TRAIL has been on clinical trial before , but need continuous infusion for several days and did not continue .Fully humanized  Mab to DR4 or  DR5 Have been developed by several companies and has been on clinical trial.   These trial will be summarized on a separate manuscript

Minor:

There seems to be a mistake in the abbreviations: ADI-PEG20 appears twice. This has been corrected